# B_3_Al_4_^+^: A Three-Dimensional Molecular Reuleaux Triangle

**DOI:** 10.3390/molecules27217407

**Published:** 2022-11-01

**Authors:** Li-Xia Bai, Mesías Orozco-Ic, Ximena Zarate, Dage Sundholm, Sudip Pan, Jin-Chang Guo, Gabriel Merino

**Affiliations:** 1Nanocluster Laboratory, Institute of Molecular Science, Shanxi University, Taiyuan 030006, China; 2Department of Chemistry, University of Helsinki, A. I. Virtasen Aukio 1, P.O. Box 55, FIN-00014 Helsinki, Finland; 3Instituto de Ciencias Químicas Aplicadas, Facultad de Ingeniería, Universidad Autónoma de Chile, Av. Pedro de Valdivia 425, Santiago 7500912, Chile; 4Fachbereich Chemie, Philipps-Universitt Marburg Hans-Meerwein-Straße, 35043 Marburg, Germany; 5Departamento de Física Aplicada, Centro de Investigación y de Estudios Avanzados, Unidad Mérida. Km 6 Antigua Carretera a Progreso. Apdo., Postal 73, Cordemex, Merida 97310, Mexico

**Keywords:** boron clusters, fluxionality, aromaticity

## Abstract

We systematically explore the potential energy surface of the B_3_Al_4_^+^ combination of atoms. The putative global minimum corresponds to a structure formed by an Al_4_ square facing a B_3_ triangle. Interestingly, the dynamical behavior can be described as a Reuleaux molecular triangle since it involves the rotation of the B_3_ triangle at the top of the Al_4_ square. The molecular dynamics simulations, corroborating with the very small rotational barriers of the B_3_ triangle, show its nearly free rotation on the Al_4_ ring, confirming the fluxional character of the cluster. Moreover, while the chemical bonding analysis suggests that the multicenter interaction between the two fragments determines its fluxionality, the magnetic response analysis reveals this cluster as a true and fully three-dimensional aromatic system.

## 1. Introduction

In 2010, Wang and coworkers detected an anionic cluster of nineteen boron atoms in the gas phase through a photoelectron spectroscopy experiment [1]. The structure has a planar pentagonal inner B_6_ core encircled by a B_13_ ring (Figure 1a). A couple of years later, Merino and coworkers found that the inner B_6_ fragment rotates almost freely with respect to the periphery [2], so they labeled this type of system as a Wankel-type rotor [2,3,4,5,6,7,8]. Over time, other boron clusters were also reported to exhibit similar behavior, including B_13_^+^ as the most prominent case [9,10]. In fact, the dynamic behavior of B_13_^+^ was experimentally verified by Asmis et al. in 2016 using cryogenic ion vibrational spectroscopy [11,12]. The main reason for this fascinating fluxionality is attributed to the multicenter bonds, often found in boron clusters [13]. Moreover, this property is not exclusive to only planar forms; compasses [14,15], drums [16], stirrers, sphere-shaped clusters [17], and other doped boron architectures have similar dynamic behavior [18,19,20,21]. Other related examples are the global minima of B_7_M_2_ and B_8_M_2_ (M = Zn, Cd, Hg), which can be described as an M_2_ dimer spinning freely on a boron wheel resembling a magnetic stirrer placed on a baseplate [22]. The reader interested in more details on fluxionality in boron clusters is referred to [12].

Inspired by these fluxional boron systems, we analyze the dynamical properties of the global minimum of B_3_Al_4_^+^ (Figure 1b). This cluster has an Al_4_ square facing a B_3_ triangle. Although this system was recently reported by Wen et al. [23], they overlooked its fluxional properties. Moreover, upon further exploration of the nature of the bonding and aromaticity, we find that this system has all the characteristics to define it as a true and fully three-dimensional aromatic system.

## 2. Computational Details

The PES was systematically explored using the Coalescence Kick (CK) program and a modified genetic algorithm implemented in GLOMOS [24,25]. Initial screening in singlet and triplet states was performed for both programs at the PBE0/LAN2DZ level [26,27]. In the range of 30 kcal mol^−1^ above the putative global minimum, the lowest isomers were minimized and characterized at the PBE0/def2-TZVP level [28]. Final energies were computed at the CCSD(T) [29]/def2-TZVP level, including the zero-point energy correction (ZPE) at the PBE0/def2-TZVP level. Thus, the energy discussion is based on the CCSD(T)/def2-TZVP//PBE0/def2-TZVP results. Natural bond orbital analysis was conducted at the PBE0 level to obtain natural atomic charges (using the NBO6 program [30]) and Wiberg bond indices (WBI) [31]. The Born–Oppenheimer molecular dynamics [32] (BOMD) were carried out at the PBE0/6-31G(d) level to ascertain the fluxional behavior. The aromatic character was evaluated by computing the magnetic response to a uniform external magnetic field. This is achieved by calculating the magnetically induced current density [33,34,35,36] (**J**^ind^) and the induced magnetic field [37,38,39] (**B**^ind^) using the GIMIC [33,34,35,36] and Aromagnetic [40] programs, respectively. Since magnetic properties computed with the BHandHLYP [41] functional yield results in good agreement with those obtained by CCSD(T) computations [42], **J**^ind^ and **B**^ind^ were calculated at the BHandHLYP/def2-TZVP level using gauge-including atomic orbitals (GIAOs) [43,44]. All these calculations were performed with Gaussian 16 [45]. Furthermore, the adaptive natural density partitioning [46] (AdNDP) analysis was also carried out to understand the nature of the interactions using the Multiwfn program [47].

## 3. Discussion

The putative global minimum of B_3_Al_4_^+^ is a singlet with *C_s_* symmetry (**1**), in which an Al_4_ square interacts with a B_3_ triangle, forming a polyhedron (see Figure 2). The B-B distances range from 1.60–1.63 Å, creating an isosceles triangle with a B-B bond aligned with an Al-Al bond. On the other hand, the Al_4_ ring is a trapezoid or quasi-square, with Al-Al bond distances lying between 2.60–2.77 Å. The distance between the two fragments is 1.73 Å.

The energetically closest isomer is only 5.1 kcal mol^−1^ above the global minimum and consists of an Al_4_ tetrahedron. The third isomer differs from the putative global minimum by exchanging a B atom for an Al atom and is 8.1 kcal/mol higher in energy. Note that all isomers in Appendix A have a B_3_ unit. As shown in Appendix A, the lowest vibrational frequency of **1** is 69 cm^−1^, which corresponds to the rotation of the B_3_ fragment. Following this smooth rotation mode, we identify the transition state (TS) belonging to the *C_s_* point group. The structural variations between the TS and the global minimum are negligible. The rotation barrier is only 0.1 kcal/mol (including the zero-point energy correction), which means that the B_3_ ring can undergo almost free rotation on top of the Al_4_ ring.

Figure 3 shows the structural evolution of B_3_Al_4_^+^ with the B_3_ triangle rotating clockwise. Starting from the global minimum, fragment B_3_ must rotate 15° to reach the maximum of the rotation barrier. A new minimum is achieved by rotating B_3_ another 15°, and a complete rotation lap is obtained by repeating this process twelve times. BOMD simulations at 300 K for 25 ps and starting from the global minimum structure show that the B_3_ triangle rotates with respect to the Al_4_ ring, just like a three-dimensional (3D) molecular Reuleaux triangle. The average distance between the B_3_ triangle and the Al_4_ quasi-square remains constant (see Appendix A).

A chemical bonding analysis is crucial to understanding the dynamical behavior of B_3_Al_4_^+^. The natural atomic charges and WBIs for **1** and TS are shown in Appendix A. The charge distribution is such that the B_3_ fragment has a charge of −2.0 |e| and the Al_4_ moiety of 3.0|e|. Instead, the WBI values show that the B-B bonds (WBI around 1.1) have a higher covalent character than the Al-Al bonds (WBI between 0.32 and 0.65), but even more fascinating is that the WBI values for the B-Al bonds are of the same magnitude as the latter (0.49 and 0.62), indicating that there is significant orbital overlap (an electron delocalization) between both fragments. We also employed AdNDP, an extension of the NBO analysis. AdNDP analysis recovers not only Lewis bond elements (lone pairs and 2c-2e bonds) but also multicenter bonds (*n*c-2e, *n* ≥ 3). For B_3_Al_4_^+^, the analysis identifies three 3c-2e σ-bonds with occupation numbers (ON) of 1.99–1.83 |e| and one π-bond (ON = 1.71 |e|) located mainly throughout the B_3_ fragment. This electron distribution is like that reported for the aromatic cyclopropenyl cation (C_3_H_3_^+^). On the other hand, the Al_4_ ring has three 4c-2e σ-bonds with ON of 1.99–1.88 |e|. The B_3_ triangle and the Al_4_ trapezoid interact via three 7c-2e bonds (Figure 4). Thus, the absence of 2c-2e bonds between the two fragments favors free rotation [48]. The bonding pattern for the TS is similar to that of the global minimum, as shown in Appendix A. The orbital compositions of canonical molecular orbitals (CMOs) for **1** and TS are listed in Appendix A, which support the bonding pattern provided by AdNDP.

As shown in Appendix A, the B_3_ ring has six σ- and two π-electrons, i.e., in principle, it satisfies Hückel’s aromaticity rule for both the σ- and π-clouds. Now, while the Al_4_ ring has six electrons, which we could classify as σ, the region between the two rings is also connected via six electrons, so both the Al_4_ fragment and the area between the rings satisfy the Hückel’s rule. The reality is that the orbital separation is complicated, and all the bonds in B_3_Al_4_^+^ have a multicentric character, i.e., they are fully delocalized in the small cage. 

Aromaticity of B_3_Al_4_^+^ is confirmed by its magnetic response to an external magnetic field perpendicular to the Al_4_ ring. The **B**^ind^ analysis reveals strong shielding values (<−50 ppm) of the *z*-component of **B**^ind^ (*B*^ind^*_z_*) along the region between the Al_4_ and B_3_ rings. This can be explained by the presence of an entirely diatropic current density in this region (Figure 5a). Thus, the interaction of both rings creates a ring current in the region between the two rings instead of being two separate ring currents, implying that B_3_Al_4_^+^ has true three-dimensional aromaticity (Figure 5b). Integration of **J**^ind^ in a plane that intersects the Al-Al and B-B bonds yields a ring–current strength (*J*^ind^) of 33.3 nA/T. Changes in the current density can be obtained by calculating the spatial derivative of the ring–current strength along the vertical *z*-axis (d*J*^ind^/d*z*) [49]. This indicates that there are no paratropic contributions along the *z*-profile. Additionally, the ring–current profile shows that the strongest current density flux occurs just below the boron triangle, confirming the three-dimensional delocalization along the *z*-axis (Appendix A). Thus, B_3_Al_4_^+^ is a true 3D aromatic fluxional cluster.

## 4. Summary

After systematically exploring the potential energy surface of clusters with formula B_3_Al_4_^+^, we confirm that the putative global minimum corresponds to a structure formed by an Al_4_ quasi-square facing a B_3_ triangle. Most interestingly, the global minimum of B_3_Al_4_^+^ exhibits fluxionality. However, its dynamic behavior involves the rotation of the B_3_ triangle on the Al_4_ ring, embodying a molecular Reuleaux triangle. Corroborated by BOMD, a complete rotation cycle includes twelve repeating steps comprising the global minimum and a transition state separated by only 0.1 kcal/mol, indicating that the rotation between both fragments is virtually free. Furthermore, the AdNDP analysis suggests that the interaction between the B and Al fragments is via three 7c-2e bonds, favoring such dynamical behavior. Regarding aromaticity, the analysis based on the magnetic response reveals a strong shielding along the *z*-axis due to an entirely diatropic current density flowing between fragments leading to a three-dimensional double (σ + π)-aromaticity. So, we find a peculiar three-dimensional fluxional 3D aromatic cluster that resembles a molecular Reuleaux triangle.

## Figures and Tables

**Figure 1 molecules-27-07407-f001:**
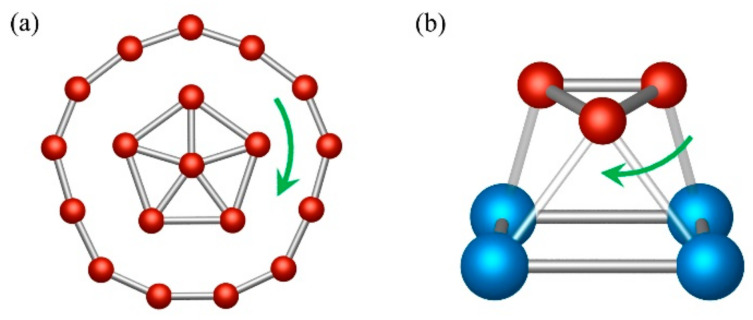
(**a**) The Wankel rotor B_19_^−^ and (**b**) Three-dimensional molecular Reuleaux triangle B_3_Al_4_^+^.

**Figure 2 molecules-27-07407-f002:**
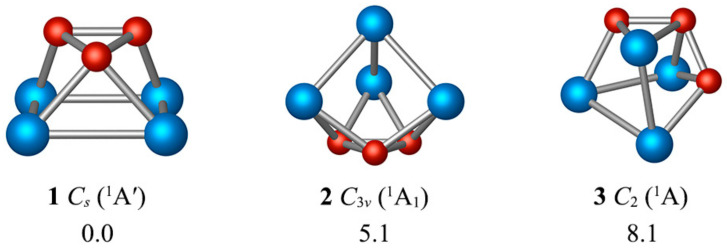
PBE0/def2-TZVP structures of the three lowest-lying energy isomers of B_3_Al_4_^+^.

**Figure 3 molecules-27-07407-f003:**
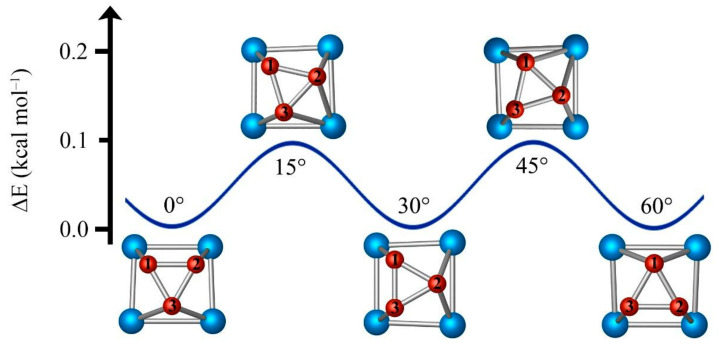
Structural evolution of three-dimensional molecular Reuleaux triangle B_3_Al_4_^+^ during the dynamic rotation.

**Figure 4 molecules-27-07407-f004:**
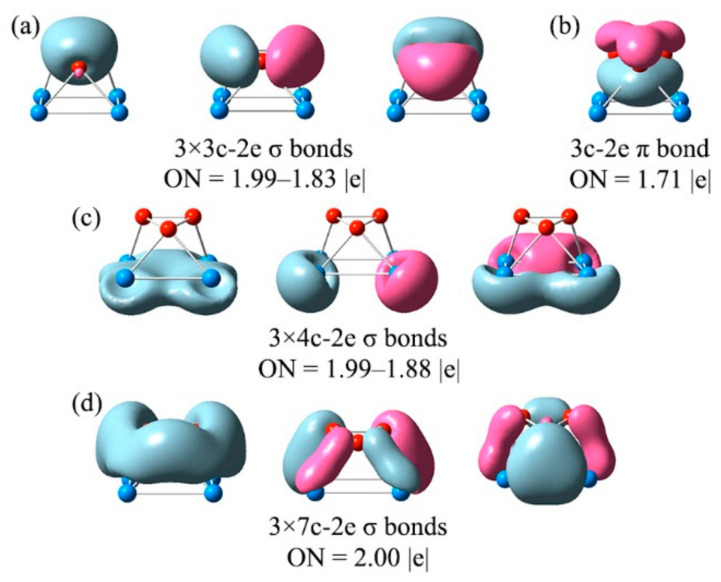
The adaptive natural density partitioning (AdNDP) bonding pattern for B_3_Al_4_^+^ (GM). ON is the occupation number.

**Figure 5 molecules-27-07407-f005:**
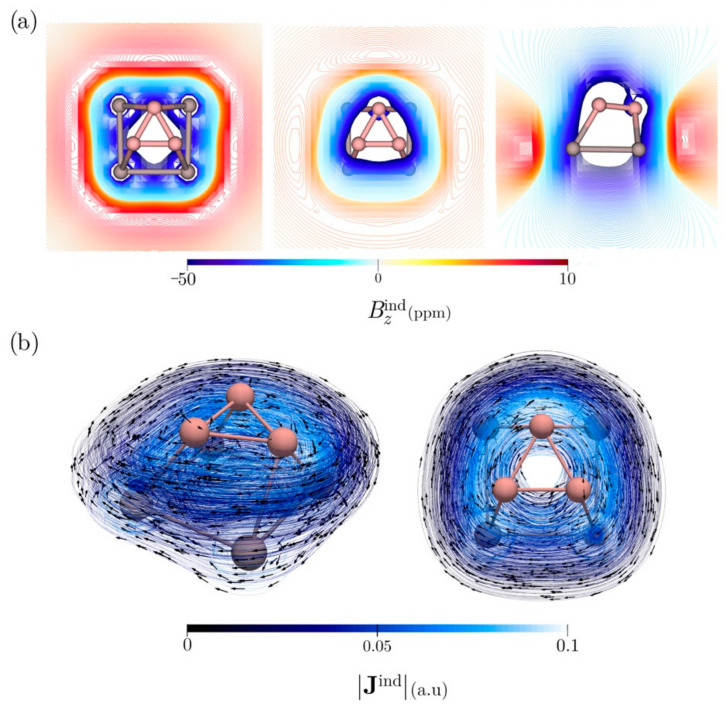
(**a**) *B*^ind^*_z_* isolines plotted in the plane of the Al_4_ framework (**left**), the B triangle (**middle**), and in a transverse plane (**right**) to B_3_Al_4_^+^. (**b**) **J**^ind^ vector maps plotted near B_3_Al_4_^+^. The arrows indicate the direction of the current density. The |**J**^ind^| scale is in atomic units (1 au = 100.63 nA/T/Å^2^). The external magnetic field is oriented parallel to the *z*-axis, perpendicular to the Al_4_ plane.

## Data Availability

Not applicable.

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
