# Peer review of "B3Al4+: A Three-Dimensional Molecular Reuleaux Triangle"

_molecules, 2022, doi:10.3390/molecules27217407_

Round 1

Reviewer 1 Report

This manuscript by Bai et al. investigates the bonding in a B3Al4+ compound by computational methods. The authors have nicely taken into account the possibility of different isomers and calculated the global minimum structure along with one transition state. The calculations and analyses for the optimized structure extend to investigations of bonding between the fragments and aromaticity. The manuscript reads well and I only have few comments and questions:

1. The authors don't have any reference compounds or compare their results with literature. This would be highly recommended.

2. Discussion section starts with journal guidelines: please delete this. Same is observed for sections Patents and Supplementary information on page 5.

3. The Figure S1 is referred a lot in the main text, maybe the three structures could be moved from the SI to the main text?

4. Figure S2 has the lowest frequency as 68.5 cm-1 and not 56 cm-1 as stated in the text.

5. In Figure 2, could the atoms be labelled so that it would be easier to see where the atoms move?  

Author Response

Reviewer #1

This manuscript by Bai et al. investigates the bonding in a B3Al4+ compound by computational methods. The authors have nicely taken into account the possibility of different isomers and calculated the global minimum structure along with one transition state. The calculations and analyses for the optimized structure extend to investigations of bonding between the fragments and aromaticity.

Reply: Thank you for your kind words.

The manuscript reads well and I only have few comments and questions:

The authors don't have any reference compounds or compare their results with literature. This would be highly recommended.

Reply: The structure and dynamical behavior of B3Al4+ are unique. To the best of our knowledge, there is not a manuscript discussing these two qualities of similar composition. Comparing it with other fragments is also not feasible since it seems that the charge distribution is +3 for the Al fragment and -2 for the B fragment, but the structure of these fragments is very different, e.g. Al4(3+) dissociates.

Discussion section starts with journal guidelines: please delete this. Same is observed for sections Patents and Supplementary information on page 5.

Reply: So sorry. It is fixed.

The Figure S1 is referred a lot in the main text, maybe the three structures could be moved from the SI to the main text?

Reply: What the reviewer states is true; we have referred to this figure several times. To prevent any misunderstanding, we have included a new figure (Figure 1) with the three lowest-lying energy structures.

Figure S2 has the lowest frequency as 68.5 cm-1 and not 56 cm-1 as stated in the text.

Reply: The correct value is 69 cm-1. It is fixed.

In Figure 2, could the atoms be labelled so that it would be easier to see where the atoms move?  

Reply: We appreciate your suggestion. In order to make the rotation easier to see, we have labeled the atoms.

Reviewer 2 Report

The ms. is well-written and the selected methods seem appropriate.

Rotational fluxionality is an intriguing curiosity. The impact would be greater if the authors would highlight why this is a significant phenomenon. What is the quantitative influence on the entropy (compared to neglecting fluxionality) for instance? Are there other areas, pure or applied, where the characterizing the behavior is important?

Do the very small energy barriers of 0.1 kcal/mol included zero-point vibrational energy? If not, then the shape of the surface in Fig. 2 could change qualitatively and the maxima might disappear.

There is leftover test at lines 90-93 and 242-243 that should be deleted.

Author Response

The ms. is well-written and the selected methods seem appropriate.

Rotational fluxionality is an intriguing curiosity. The impact would be greater if the authors would highlight why this is a significant phenomenon. What is the quantitative influence on the entropy (compared to neglecting fluxionality) for instance? Are there other areas, pure or applied, where the characterizing the behavior is important?

Reply: This is an interesting point. We recently summarized this type of fluxionality in an article in Accounts of Chemical Research. We have added the following sentence in the Introduction: "The reader interested in more details on fluxionality in boron clusters is referred to Ref. 12”

Do the very small energy barriers of 0.1 kcal/mol included zero-point vibrational energy? If not, then the shape of the surface in Fig. 2 could change qualitatively and the maxima might disappear.

Reply: Barriers includes zero-point energy corrections. The above is highlighted in the computational details “Final energies were computed at the CCSD(T)29/def2-TZVP level, including the zero-point energy correction (ZPE) at the PBE0/def2-TZVP level.”

There is leftover test at lines 90-93 and 242-243 that should be deleted.

Reply: So sorry. It is fixed.

Reviewer 3 Report

Dear Authors,

I have enjoyed reading the manuscript titled as 'B3Al4+: A Three -Dimensional Molecular Reuleaux Triangle.' I think it will be a very interesting reading for the audience of the journal Molecules. The methodologies selected for this scientific work is sound in nature and the conclusions are documented in a lucid manner. Also, the experimental community should find this article very intriguing because of the aromatic characteristics of the species. 

I propose the following minor corrections of the manuscript-

1. Line 59-64 is a duplicate of line 41-46

2. In line 94 (and more), by the notation '(1)', I believe the authors have indicated the first structure of Figure S1. It would be nice to clarify the indication. 

3. In line 102, it is stated that the vibrational frequency of 1 is 56 cm-1, but from Figure S2, I believe it should be 68.5 cm-1

4. I couldn't find the supplementary information II, the short movie extracted from the BOMD simulation.

Author Response

I have enjoyed reading the manuscript titled as 'B3Al4+: A Three -Dimensional Molecular Reuleaux Triangle.' I think it will be a very interesting reading for the audience of the journal Molecules. The methodologies selected for this scientific work is sound in nature and the conclusions are documented in a lucid manner. Also, the experimental community should find this article very intriguing because of the aromatic characteristics of the species.

Reply: Thank you for your kind words.

I propose the following minor corrections of the manuscript-

Line 59-64 is a duplicate of line 41-46

Reply: So sorry. It is fixed.

In line 94 (and more), by the notation '(1)', I believe the authors have indicated the first structure of Figure S1. It would be nice to clarify the indication.

Reply: To prevent any misunderstanding, we have included a new figure (Figure 1) with the three lowest-lying energy structures.

In line 102, it is stated that the vibrational frequency of 1 is 56 cm-1, but from Figure S2, I believe it should be 68.5 cm-1

Reply: The correct value is 69 cm-1. It is fixed.

I couldn't find the supplementary information II, the short movie extracted from the BOMD simulation.

Reply: It is fixed.